# Elderly Hip Osteoarthritis: A Review of Short-Term Pain Relief Through Non-Weight-Bearing Therapies

**DOI:** 10.3390/jfmk10020124

**Published:** 2025-04-08

**Authors:** Olivia Norato, Sarah Velez, Arbonor Lleshi, Gordon Lam, Marlon Morales, Glory Udechi, Edwin Cung, Jean-Philippe Berteau

**Affiliations:** 1Department of Physical Therapy, City University of New York-College of Staten Island, Staten Island, NY 10314, USAmarlon.morales62@bcmail.cuny.edu (M.M.);; 2New York Center for Biomedical Engineering, City University of New York-City College of New York, New York, NY 10031, USA; 3Nanoscience Initiative, Advanced Science Research Center, City University of New York, New York, NY 10031, USA

**Keywords:** manual therapy, aquatic therapy, hip osteoarthritis, pain relief

## Abstract

Older individuals with hip osteoarthritis (OA) who have difficulty walking, climbing stairs, or performing daily tasks often find non-weight-bearing (NWB) exercises essential for rebuilding strength and preserving function without further stressing the joints. In addition, those with a higher body mass index (BMI) particularly benefit from NWB therapy, as it alleviates joint pressure while facilitating safe and effective rehabilitation. Thus, NWB interventions, such as manual therapy (MT) and aquatic therapy (AT), are especially critical for older adults aged 60 and above, offering pain relief and functional improvement by minimizing gravitational impact on the hip joint. This review examines the effectiveness of these approaches in managing hip OA symptoms and decreasing pain. The inclusion criteria for the study consisted of randomized controlled trials or controlled trials focused on adult patients with primary osteoarthritis of the hip joint, utilizing interventions such as MT (including thrust joint mobilizations, non-thrust/oscillatory mobilizations, and soft tissue mobilization) or AT (including hydrotherapy and water therapy), and assessing outcomes related to pain. We selected nine studies that included a total of *n* = 1037 individuals. It evaluated outcomes such as self-reported pain levels using measures like the Western Ontario and McMaster Universities Osteoarthritis (WOMAC), Numeric Rating Scale (NRS), and Visual Analog Scale (VAS). Beyond statistical differences, both therapies were evaluated for Minimal Clinically Important Difference (MCID). While MT studies indicated a decrease in pain according to pain index scores, they showed short-term effectiveness till five weeks but lacked sustained clinical efficacy beyond this period. AT showed positive results within a ten-week period, although its effectiveness seemed to level off beyond this duration, falling below the threshold of clinical efficiency. After 10 weeks of treatment, there is no discernible clinical benefit in terms of pain reduction. Both interventions without gravitational impact seem suitable for providing short-term pain relief for primary hip osteoarthritis patients, but long-term pain relief—meaning after ten weeks—should be maintained through therapeutic exercise and patient education.

## 1. Introduction

Osteoarthritis (OA) is a prevalent joint disorder, affecting over 240 million people worldwide and more than 32 million in the United States alone [1,2,3]. It primarily involves pathological changes in connective tissue, articular cartilage, and joint surfaces, commonly afflicting the hips, knees, hands, and spine [4,5,6,7]. While primary OA results from chronic degeneration due to overuse and increased loading, secondary OA stems from identifiable causes such as trauma, surgery, infection, and necrosis. The consequences of OA, particularly in the hips, are profound, leading to pain, stiffness, and reduced flexibility, significantly impacting daily function and quality of life [5,8,9,10]. Among individuals aged 60 to 90, the prevalence of primary hip osteoarthritis (HOA) is estimated at 8.0% for women and 6.7% for men [11]. Risk factors include prior joint trauma, malalignment, and comorbidities such as obesity, hypertension, cardiovascular disease, and diabetes [12].

The pathophysiological process of HOA involves the progressive degradation of articulation surfaces because of abnormal loading or shear forces, starting with molecular derangement and leading to cartilage degradation, bone remodeling, osteophyte formation, joint inflammation, and loss of function [13] associated with pain [6]. Progression through HOA’s four-stage severity grading system—from early to severe—results in varying levels of asymmetry or decreased synovial joint space, evident in radiographic imaging [14,15]. As OA primarily affects older adults, the breakdown of cartilage in the joints leads to decreased joint space and increased friction, resulting in pain and discomfort. Aging and obesity, along with cellular and matrix changes in joint tissues, contribute significantly to OA development, with the hip joint being particularly susceptible [16,17,18]. In the context of global osteoarthritis (OA) management, non-pharmaceutical interventions play a critical role in addressing pain and functional limitations, complementing pharmacological and surgical approaches. While modalities intervention in Physical Therapy such as transcutaneous electrical nerve stimulation (TENS), continuous ultrasound, and optional deep microwave diathermy have demonstrated efficacy in achieving clinically significant pain relief, reducing scores by 20% [5,7,19,20], some patients are not eligible for these interventions and resources are not always accessible. While exercise has been the gold standard in Physical therapy intervention as it shows strong efficiency [21,22,23,24,25], patients experiencing significant pain, stiffness, or joint instability may find weight-bearing activities intolerable. Thus, these patients who might have difficulty walking, climbing stairs, or performing daily activities find non-weight-bearing (NWB) exercises particularly beneficial for rebuilding strength and preserving function without risking further joint damage. Those with a high body mass index (BMI) also gain significant advantages from NWB therapy, as it reduces the mechanical stress on the hip joint. Excess body weight amplifies the force exerted on the hip by three to six times the individual’s body weight during activities like walking, accelerating cartilage deterioration. In addition to this, weight loss has been recognized as a crucial component in managing osteoarthritis, particularly hip OA, as it reduces mechanical stress on the joints and decreases systemic inflammation, potentially alleviating symptoms and improving overall joint function [26]. NWB exercises allow these patients to safely strengthen muscles, improve joint range of motion, and maintain mobility while minimizing pain and strain on the affected joints. As a result, non-weight-bearing therapy is particularly vital for individuals with osteoarthritis (OA), especially those who are overweight, as it reduces joint stress while supporting functional recovery. In managing hip OA, therapeutic strategies focus on minimizing uneven joint stress and improving mobility without the impact of gravitational forces. Non-weight-bearing interventions, such as manual therapy (MT) and aquatic therapy (AT), offer targeted solutions by improving joint mobility while avoiding gravitational stress. [27,28,29,30] (Figure 1). These interventions can relax tissues, enhance range of motion, mobilize and manipulate joints, stabilize them, modulate pain responses, and reduce joint inflammation. Conversely, AT involves performing exercises in water, offering a low-impact activity that alleviates joint pain and enhances mobility [31]. These approaches are especially critical for individuals with OA who are overweight, though specific protocols integrating MT and AT for hip OA remain lacking. This highlights the need for further research and standardized treatment strategies in the global management of OA [7].

This narrative review aims to assess pain relief for hip osteoarthritis in elderly individuals without gravitational impact using manual therapy and aquatic therapy among individuals aged 60 years and above. By synthesizing the existing literature and evaluating outcomes such as self-reported pain levels measured through established scales such as the Western Ontario and McMaster Universities Osteoarthritis (WOMAC), Numeric Rating Scale (NRS), and Visual Analog Scale (VAS), this review seeks to provide valuable insights into optimal approaches for managing HOA pain and enhancing the quality of life for affected individuals. Beyond statistical differences, both therapies were evaluated for Minimal Clinically Important Difference (MCID) [32,33,34].

## 2. Materials and Methods

For this narrative review, we created a PICO that includes our population, intervention, comparison, and outcome. For populations of patients aged 60 years and over with hip OA, the intervention included aquatic exercise and manual therapy; we did not include a specific comparison, and the outcome measured changes in pain measured during aquatic exercise using either Visual analog scores (VASs) or The Western Ontario and McMaster Universities Osteoarthritis Index (WOMAC). We evaluated articles using three different online databases (PubMed, EBSCO, and Cochrane) using the following MeSH words: [Hip] OR [acetabulum] OR [coxal] AND [OA] OR [osteoarthritis] OR [arthritis] OR [joint degeneration] OR [joint wear tear] OR [degenerative joint disease] OR [cartilage degeneration] OR [arthrosis] OR [coxarthrosis] OR [degenerative disease] OR [joint disease] AND [water therapy] OR [aquatic therapy] OR [hydrotherapy] OR [aquatic exercise] OR [water exercise] OR [soft tissue mobilization] OR [massage] OR [mobilization] OR [manipulation] OR [myofascial release] AND [VAS] OR [WOMAC] OR [pain] between September 2024 and December 2024. We followed the Scale for the Assessment of Narrative Review Articles—SANRA—guidelines to perform this narrative review.

Regarding manual therapy, we obtained 971 articles in total: 371 from PubMed, 123 from CINAHL, and 477 from Cochrane. We removed 133 duplicates, to be left with 838 articles. We then screened these articles using defined inclusion and exclusion criteria (Table A1). A total of 831 articles were excluded based on these criteria, and 2 articles were sought but not retrieved, leaving 5 articles to be appraised. Regarding Aquatic therapy, we gathered 95 articles from PubMed, 29 articles from EBSCO, and 79 from Cochrane for a total of 203 articles. After deleting 45 duplicates a total of 158 articles were left. Then, we screened the remaining articles using our inclusion and exclusion criteria. With these criteria, 118 articles were removed, leaving 40 articles to be assessed for eligibility. After assessing the eligible articles, 27 studies were studied to be included in the review. Another round of assessment excluded 22 reviews leaving 5 reports of included studies.

A relative risk of bias assessment was conducted for articles including a WOMAC or VAS pain score. The relative risk reduction (RRR) was also calculated as RRR = risk of outcome in the treatment group/risk of outcome in the control group. Regarding manual therapy and aquatic therapy, Table 1 includes a summary of each article with the relative risk reduction. Figure 2 plots the data of RR regarding the duration of intervention. RRR is a concept used to measure how much an intervention reduces a particular outcome, such as pain levels measured by the Visual Analog Scale (VAS). In the context of VAS scores, RRR helps us understand the proportion of pain reduction achieved through an intervention like manual therapy or aquatic therapy. To determine the RRR, we compare the initial VAS score before the intervention with the VAS score after the intervention. The difference between these two scores shows the absolute reduction in pain. The RRR then expresses this difference as a percentage of the initial VAS score. This percentage tells us how effective the intervention was in reducing pain relative to the starting point.

## 3. Results

Table 1 displays the results of our analysis based on the articles we chose for this systematic review. We selected nine studies that included a total of n = 1037 individuals. The main takeaway from the chart is the relative risk and clinical efficiency of each study. Furthermore, Figure 2 shows the relationship between pain reduction based on manual therapy and how long the treatment was. Hoeksma et al. [22] reported a 35.2% reduction in pain using VAS for a group of 112 individuals undergoing MT. Abbott et al. [21] and Beselga et al. [35] (observed 11.8% and 32.5% pain reduction, respectively, using NRS in their MT studies. French et al. [36] found a lower 6.4% reduction with NRS for MT. In AT studies, Foley et al. [37] reported a 20% reduction in WOMAC pain for 105 participants, while Hinman et al. [38] found a 28% reduction in WOMAC and 20% in VAS for 71 participants. Cochrane et al. [39] showed varying pain reductions in WOMAC over 6, 12, and 18 months (8%, 9.5%, and 4.4%, respectively) for a group of 106. Fransen et al. [40] achieved a notable 46.5% pain reduction in WOMAC for a large group of 152 using AT combined with Tai Chi. Finally, Wang et al. [31] reported 6.3% and 20.8% reductions in VAS for 42 participants over 6 and 12 weeks of aquatic exercise, respectively. These findings underscore that both therapies are effective for short-term pain relief, with AT potentially offering more substantial improvements, especially when combined with other exercises like Tai Chi. Both the type of therapy and the duration are crucial for achieving clinical efficiency, with longer and combined therapy approaches potentially offering better outcomes, although shorter interventions can also be effective for certain therapies.

## 4. Discussion

These studies collectively underscore the potential benefits of manual therapy and exercise therapy in managing OA. Manual therapy (MT) demonstrated a pain reduction range of 6.4% to 35.2%, with the highest improvement seen at 35.2%, while aquatic therapy (AT) ranged from 4.4% to 46.5%, with the most substantial reduction of 46.5% when combined with Tai Chi. When ranking the therapies by effectiveness, AT combined with Tai Chi achieved the greatest pain relief, followed by MT and then standard AT. The minimum reductions were observed in MT at 6.4% and AT at 4.4%, showing that while both therapies provide pain relief, the results can vary significantly depending on the method and duration. The duration of pain improvement varied significantly across the different therapies and study durations. Manual therapy typically showed short-term benefits, with notable reductions in pain observed within a few weeks, such as a 35.2% reduction over 5 weeks, and immediate improvements, like the 32.5% reduction reported with no treatment duration. In contrast, aquatic therapy demonstrated more variability, with larger improvements seen in the short term (20% to 28% reduction over 6 weeks) but more modest reductions in longer durations (4.4% to 9.5% over 24 to 72 weeks). Combining aquatic therapy with Tai Chi yielded the most substantial improvement, with a 46.5% reduction over 12 weeks. Overall, shorter interventions generally produced more significant pain relief, while longer therapies led to more gradual and sometimes less pronounced benefits. MT demonstrates efficacy in pain reduction, stiffness alleviation, and functional improvement, with sustained effects over time. Exercise therapy, both alone and in combination with MT, shows promise in enhancing physical function and range of motion. While manual therapy studies indicated a decrease in pain according to pain index scores, manual therapy showed short-term effectiveness till five weeks but lacked sustained clinical efficacy beyond this period. Aquatic therapy showed positive results within a ten-week period, although its effectiveness seemed to level off beyond this duration, falling below the threshold of clinical efficiency. Both seem suitable for providing short-term pain relief for primary hip osteoarthritis patients, but long-term pain relief—meaning after ten weeks—should be maintained through therapeutic exercise and patient education.

Regarding manual therapy, the collective findings of multiple studies shed light on the efficacy of various therapies in managing osteoarthritis (OA), particularly in the context of manual therapy associated with or in comparison to exercise therapy. Hoeksma et al. [22] conducted a study involving 109 OA patients, comparing manual therapy with exercise therapy. Both groups exhibited baseline comparability in demographics and OA severity. However, attrition rates were notable, with a gradual loss of participants over follow-up. Manual therapy showed higher success rates at 5 weeks compared to exercise therapy, with favorable outcomes in pain, stiffness, and range of motion. Long-term analysis revealed sustained improvement in manual therapy, albeit with declining effects. Abbott et al. [21] similarly examined manual therapy alongside usual care and exercise therapy in 206 participants. They found significant pain reductions favoring manual therapy, with exercise therapy also showing promise. Notably, antagonistic interactions between manual and exercise therapies were observed. Beselga et al. [35] focused on Mulligan’s Mobilization with Movement (MWM) in 40 participants, reporting significant improvements in hip pain intensity, range of motion, and functional tests compared to a sham group. French et al. [36] explored exercise therapy alone and in combination with manual therapy in 131 participants. While dropout rates were modest, patient satisfaction was higher in the combined therapy group. Significant improvements in physical function and range of motion were observed in both treatment groups compared to the control. Wright et al. [30] investigated correlational analyses and found higher correlations in the manual therapy group, though none reached significance. The comparison of regression slopes suggested distinct interactions between within-session findings and targeted outcomes in manual therapy versus supervised neglect.

Regarding aquatic therapy in managing osteoarthritis (OA) symptoms and after systematically reviewing the available studies, it was found that aquatic therapy is effective for short-term pain relief in patients with hip osteoarthritis (OA), with benefits evident within the first four months of treatment. However, its effectiveness plateaus over time, falling below clinical efficiency. The Foley study [37] found that hydrotherapy improved physical function but not long-term muscle strength compared to gym-based exercises due to differences in muscle loading and contraction phases. The Hinman study [38] linked success in water-based programs to adherence, which might not be sustained in clinical settings, potentially due to a lack of equipment. Limitations included inconsistent measurement scales, varied program lengths, and a predominant focus on knee rather than hip OA, making direct comparisons difficult. Sampling and adherence issues were also noted in the Cochrane study. The Fransen study [40] suggested that while water buoyancy alleviates pain, it also reduces muscle resistance, limiting long-term benefits. The Wang study [31] confirmed short-term improvements in joint flexibility, muscle strength, and aerobic fitness but indicated that moderate aquatic exercise does not increase pain, underscoring the need for further longitudinal studies to confirm long-term effectiveness. Despite these challenges, aquatic therapy was found to be safe and beneficial in the short term, but more consistent and focused research is needed to determine its long-term effectiveness and establish standardized treatment protocols.

Regarding the limitations of our work, the studies selected include notable attrition rates, short-term efficacy with benefits of manual therapy lasting only up to five weeks and aquatic therapy up to four months, and inconsistent measurement scales complicating direct comparisons. Blinding difficulties and variability in program lengths and intensities further challenge the reliability of the results. Antagonistic interactions between manual and exercise therapies were observed, and a predominant focus on knee rather than hip OA limits generalizability [5,19,20,41]. Sampling and adherence issues, along with limited long-term data on muscle strength and standardized protocols, reduce the robustness of conclusions. Additionally, heterogeneity in study populations and the lack of comprehensive longitudinal data underscore the necessity for more rigorous, consistent, and focused research to determine the long-term effectiveness and optimal application of these therapies for managing hip OA. Further research is warranted to assess the duration of pain relief provided by manual therapy and/or aquatic therapy beyond five weeks and its potential in combination with other modalities. Nonetheless, manual therapy remains a valuable short-term intervention for reducing pain in individuals with hip OA as offering immediate relief, and aquatic therapy contributes to improved functional outcomes and patient satisfaction. The variation in therapy duration in our review reflects the range of approaches documented in the current literature, where short-term interventions are often evaluated for immediate symptom relief, while longer-term studies assess sustained benefits and potential long-term efficacy. This diversity in duration allows for a comprehensive understanding of therapy effectiveness, aligning with research that highlights the importance of evaluating both immediate and prolonged outcomes in managing hip osteoarthritis. While this narrative review provides valuable insights into the effectiveness of non-weight-bearing therapies for hip osteoarthritis, it is important to recognize its qualitative nature, which allows for a broad exploration of the literature. Although narrative reviews may not capture every quantitative study, they offer a unique perspective by synthesizing diverse research findings. This approach enables us to highlight key trends and themes that might be overlooked in more systematic reviews.

The duration of pain improvement varied significantly across the different therapies, revealing notable discrepancies in outcomes. Manual therapy generally provided more immediate pain relief, with reductions ranging from 6.4% to 35.2%, suggesting that it is effective for short-term improvements. However, in longer-term studies, the reduction was much lower at 11.8%, indicating that prolonged manual therapy may yield more gradual and less significant improvements. Aquatic therapy, on the other hand, showed more variability, with larger reductions in the short term (20% to 28% over 6 weeks) but more modest gains in the longer term (4.4% to 9.5% over 24 to 72 weeks). The most substantial pain reduction was observed when aquatic therapy was combined with Tai Chi (46.5% over 12 weeks), pointing to the potential added benefit of incorporating other therapeutic exercises. These discrepancies suggest that the combination of therapies, as well as the duration, play key roles in the effectiveness of pain relief, with short-term treatments generally offering quicker, more substantial improvements, while longer interventions may not always result in proportional gains. Compared to modalities used in non-pharmaceutical treatment, manual therapy (MT) and aquatic therapy (AT) primarily offer short-term pain relief and functional improvements for osteoarthritis (OA) patients, whereas diathermy, ultrasound (US), and electrical stimulation (ES) therapies can provide both short-term and longer-lasting benefits [5,7]. MT has shown effectiveness within the first five weeks but lacks sustained efficacy beyond that period. AT offers short-term benefits as well, especially in improving physical function and reducing pain, but its effectiveness plateaus after about four months. These therapies often require continued patient engagement, including exercise and education, for long-term management. In contrast, diathermy, US, and ES are generally more effective over both short and long durations, particularly when combined with therapeutic exercises. Diathermy and US, particularly in modalities like continuous ultrasound (CU) or mixed ultrasound, have shown significant pain reduction in knee OA, with some protocols demonstrating relief over several weeks. However, while diathermy, ultrasound, and electrical stimulation offer consistent and longer-lasting results, mainly for the knee [42,43,44,45,46,47,48], manual therapy (MT) and aquatic therapy (AT) also present distinct advantages, particularly in promoting immediate relief and improving overall functional outcomes. MT, with techniques like joint mobilizations, can significantly reduce pain and stiffness in the short term, making it an excellent option for patients seeking rapid relief. Additionally, it provides individualized care tailored to the patient’s needs, fostering a strong therapeutic alliance. AT, with its low-impact nature, is particularly beneficial for patients with hip OA, offering a safe environment for improving range of motion, strength, and cardiovascular health. Both MT and AT emphasize patient engagement and education, encouraging active participation in their own recovery, which can lead to improved patient satisfaction and adherence. We also acknowledge the importance of strength work with body weight in managing hip osteoarthritis, as highlighted in the introduction. The study by Salis et al. [26] provides valuable insights into the benefits of weight loss, which can complement strength work by reducing mechanical stress on the hip joint and decreasing systemic inflammation. This study demonstrated that weight loss is dose-dependently associated with reductions in symptoms of hip osteoarthritis, supporting the notion that weight management can enhance the effectiveness of strength exercises. Together, these therapies offer valuable complementary options for managing OA, especially when combined with other interventions like exercise, for a holistic approach to pain management and improved quality of life.

## 5. Conclusions

This review investigates how effective manual therapy (MT) and aquatic therapy (AT) are in reducing hip osteoarthritis pain in people aged 60 and older. Both therapies were evaluated not only for statistical differences but also for their ability to meet the Minimal Clinically Important Difference (MCID). Studies on manual therapy showed a decrease in pain scores initially, with noticeable short-term effectiveness lasting up to five weeks. However, there was no sustained clinical effectiveness beyond this timeframe. Aquatic therapy demonstrated positive results within a ten-week period, but its effectiveness seemed to level off afterward, dropping below the threshold considered clinically effective. After 10 weeks of treatment, there is no discernible clinical benefit in terms of pain reduction. While both therapies provide short-term pain relief for individuals with primary hip osteoarthritis, managing pain over the long term beyond ten weeks should involve strategies such as therapeutic exercise and patient education. Thus, manual therapy (MT) and aquatic therapy (AT) offer valuable benefits in pain relief, functional improvement, and patient satisfaction, particularly through individualized care and low-impact environments. When combined with other interventions, MT and AT can significantly enhance patient outcomes, promoting long-term mobility and overall well-being. To summarize, this review highlights the potential of manual therapy and aquatic therapy as effective non-weight-bearing interventions for short-term pain relief in elderly individuals with hip osteoarthritis. While both therapies demonstrate promising results, long-term pain management should incorporate therapeutic exercise and patient education to sustain benefits. Clinicians are encouraged to integrate these therapies into comprehensive treatment plans, tailoring them to individual patient needs and capabilities.

## Figures and Tables

**Figure 1 jfmk-10-00124-f001:**
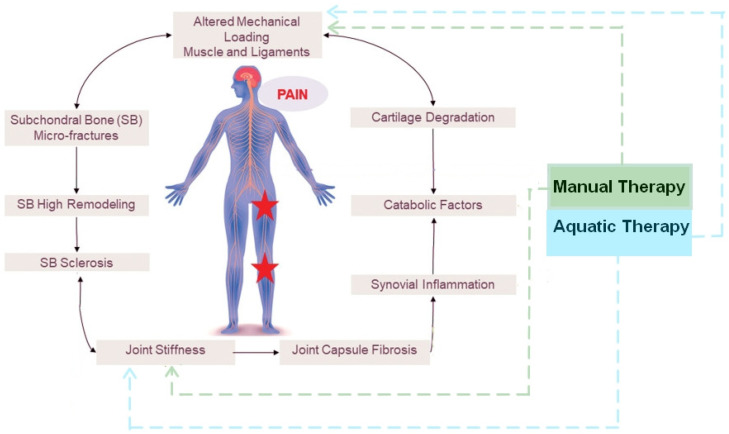
Effect of manual therapy and aquatic therapy on OA vicious circle of progression and pain mechanism associated.

**Figure 2 jfmk-10-00124-f002:**
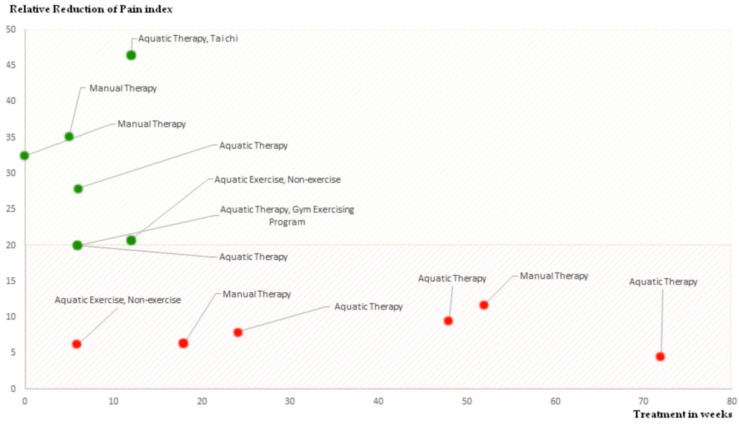
The graph illustrates the relationship between treatment duration in weeks and the reduction of the pain index; indeed, both therapies were evaluated for their ability to meet the Minimal Clinically Important Difference (MCID). The red area denotes reductions below 20%, which are considered below the threshold of clinical effectiveness. The green area represents reductions of 20% or more, meeting the minimum criteria for clinical efficiency. The graph shows that after 10 weeks of treatment, there is no discernible clinical benefit in terms of pain reduction.

**Table 1 jfmk-10-00124-t001:** Summary of studies on manual therapy (MT) and aquatic therapy (AT) for hip osteoarthritis pain relief.

Author	*n*	Intervention	Duration (in Weeks)	Outcome	RR%
Hoeksma et al. [22]	112	Manual Therapy	5	VAS	35.2
Abbott et al. [21]	105	Manual Therapy	52	NRS	11.8
Beselga et al. [35]	40	Manual Therapy	0	NRS	32.5
French et al. [36]	50	Manual Therapy	18	NRS	6.4
Foley et al. [37]	105	Aquatic Therapy and Gym Exercising Program	6	WOMAC pain	20
Hinman et al. [38]	71	Aquatic Therapy	6	WOMAC	28
			6	VAS	20
Cochrane et al. [39]	106	Aquatic Therapy	24	WOMAC pain	8
	106	Aquatic Therapy	48	WOMAC pain	9.5
	106	Aquatic Therapy	72	WOMAC pain	4.4
Fransen et al. [40]	152	Aquatic Therapy and Tai chi	12	WOMAC	46.5
Wang et al. [31]	42	Aquatic Exercise	6	VAS	6.3
	42	Aquatic Exercise	12	VAS	20.8

## Data Availability

No new data were created or analyzed in this study.

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
