# Peer review of "Elderly Hip Osteoarthritis: A Review of Short-Term Pain Relief Through Non-Weight-Bearing Therapies"

_jfmk, 2025, doi:10.3390/jfmk10020124_

Round 1
Reviewer 1 Report
Comments and Suggestions for Authors
I congratulate the authors for the article. Please find my comments below.
Abstract - comprehensive and understandable.
Introduction - comprehensive and understandable. A good literature review that describes the current state-of-the-art.
However, there are some rather outdated references that could be replaced with novel ones.
Methods - the search algorithm (as set in each database) together with the used boolean words and the timeline when the search was done should be specifically described.
Results - I believe it is important to have a flow chart of the search and selection process. I'd suggest you use a PRISMA flow chart for systematic reviews and meta-analysis.
Discussion - a final paragraph with study limitations should be added. The type of study (narrative review) and its inability to systematically include all the existing state-of-the-art should be mentioned as a major limitation. Furthermore, I am convinced that there are other limitations that authors need to address (there are no studies without limitations).
Conclusion - having in mind the important outcomes deriving from this study, I suggest adding a couple of sentences with suggestions for the implication of your findings into clinical practice.
Author Response
Thank you very much for taking the time to review this manuscript. Please find the detailed responses below and the corresponding revisions/corrections highlighted in the resubmitted files.
Questions for General Evaluation
Does the introduction provide sufficient background and include all relevant references? We appreciate the reviewer's feedback regarding the references. We have updated the introduction with more recent references to reflect the current state-of-the-art.
Are all the cited references relevant to the research? We have reviewed the references and replaced outdated ones with more recent and relevant studies.
Is the research design appropriate? The narrative review design is appropriate for synthesizing existing literature on the topic.
Are the methods adequately described? We have added a detailed description of the search algorithm, including the Boolean operators used and the timeline of the search.
Are the results clearly presented? We have clarify the search and selection process.
Are the conclusions supported by the results? The conclusions are well-supported by the findings presented in the results section.
Point-by-point response to Comments and Suggestions for Authors
Comments 1:The abstract and introduction are comprehensive, but some references are outdated.
Response 1: Thank you for pointing this out. We agree that it is important to include the most recent literature. We have updated the references in the introduction to reflect current research trends. These changes can be found in the introduction section, particularly in the discussion of the current state-of-the-art.
Berteau, J.-P. Knee Pain from Osteoarthritis: Pathogenesis, Risk Factors, and Recent Evidence on Physical Therapy Interventions. JCM 2022, 11, 3252, doi:10.3390/jcm11123252.
Yu, H.; Huang, T.; Lu, W.W.; Tong, L.; Chen, D. Osteoarthritis Pain. IJMS 2022, 23, 4642, doi:10.3390/ijms23094642.
Berteau, J.-P.P. Systematic Narrative Review of Modalities in Physiotherapy for Managing Pain in Hip and Knee Osteoarthritis: A Review. Medicine (Baltimore) 2024, 103, e38225, doi:10.1097/MD.0000000000038225.
Briggs, K.; Matheny, L.; Steadman, J. Improvement in Quality of Life with Use of an Unloader Knee Brace in Active Patients with OA: A Prospective Cohort Study. J Knee Surg 2012, 25, 417–422, doi:10.1055/s-0032-1313748.
Király, M.; Varga, Z.; Szanyó, F.; Kiss, R.; Hodosi, K.; Bender, T. Effects of Underwater Ultrasound Therapy on Pain, Inflammation, Hand Function and Quality of Life in Patients with Rheumatoid Arthritis – a Randomized Controlled Trial. Brazilian Journal of Physical Therapy 2017, 21, 199–205, doi:10.1016/j.bjpt.2017.04.002.
Yang, X.; He, H.; Ye, W.; Perry, T.A.; He, C. Effects of Pulsed Electromagnetic Field Therapy on Pain, Stiffness, Physical Function, and Quality of Life in Patients With Osteoarthritis: A Systematic Review and Meta-Analysis of Randomized Placebo-Controlled Trials. Phys Ther 2020, 100, 1118–1131, doi:10.1093/ptj/pzaa054.
Comments 2: The methods section should specifically describe the search algorithm and timeline.
Reponse 2 : We have now detailed the search algorithm used for each database, including the Boolean operators and the timeline when the search was conducted. This information has been added to the methods section. " using the following MeSH words: [Hip] OR [acetabulum] OR [coxal] AND [OA] OR [oste-oarthritis] OR [arthritis] OR [joint degeneration] OR [joint wear tear] OR [degenerative joint disease] OR [cartilage degeneration] OR [arthrosis] OR [coxarthrosis] OR [degenerative disease] OR [joint disease] AND [water therapy] OR [aquatic therapy] OR [hydrotherapy] OR [aquatic exercise] OR [water exercise] OR [soft tissue mobilization] OR [massage] OR [mobilization] OR [manipulation] OR [myofascial release] AND [VAS] OR [WOMAC] OR [pain] between September 2024 and December 2024"
Comments 3: A flow chart of the search and selection process should be included.
Response 3: We have not included a PRISMA flow chart to visually represent the search and selection process because we did not do a systematic review but a narrative review following the SANRA guidelines not the PRISMA guidelines.
Comments 4: A final paragraph with study limitations should be added to the discussion.
Response 4 : a paragraph was already there but we made it more clear to the reader. We have added a paragraph discussing the limitations of the study, including the narrative review format and its inability to systematically include all existing literature. This addition can be found at the end of the discussion section : " While this narrative review provides valuable insights into the effectiveness of non-weight-bearing therapies for hip osteoarthritis, it is important to recognize its qualitative nature, which allows for a broad exploration of the literature. Although narrative reviews may not capture every quantitative study, they offer a unique perspective by synthesizing diverse research findings. This approach enables us to highlight key trends and themes that might be overlooked in more systematic reviews."
Comments 5: Add suggestions for clinical practice in the conclusion. We have added a couple of sentences in the conclusion suggesting how the findings could be implemented in clinical practice. These additions can be found in the conclusion section : " To summarize, this review highlights the potential of manual therapy and aquatic thera-py as effective non-weight-bearing interventions for short-term pain relief in elderly indi-viduals with hip osteoarthritis. While both therapies demonstrate promising results, long-term pain management should incorporate therapeutic exercise and patient educa-tion to sustain benefits. Clinicians are encouraged to integrate these therapies into com-prehensive treatment plans, tailoring them to individual patient needs and capabilities"
Reviewer 2 Report
Comments and Suggestions for Authors
This narrative review investigates the effectiveness of manual therapy (MT) and aquatic therapy (AT) in reducing hip osteoarthritis pain in people aged 60 and older. Both therapies' effects on hip pain were evaluated in different studies screened with defined inclusion and exclusion criteria. The article demonstrates clarity and provides insights and findings that contribute to the discipline. The structure of the article is logical and well-organized, guiding the reader through each section with ease. This paper highlights the need for additional research and the impact these therapies have on clinical practices.
Author Response
We are pleased that the reviewer found the article to be a valuable contribution to the field.
Reviewer 3 Report
Comments and Suggestions for Authors
Please check the comments carefully.

Author Response
We would like to warmly thanks the reviewer for their comments and suggestions for authors, we are happy to provide a Point-by-point response to them :
Comments 1: Include a process flow diagram in the methods section.
Response 1: We have not included a PRISMA flow chart to visually represent the search and selection process because we did not do a systematic review but a narrative review following the SANRA guidelines not the PRISMA guidelines.
Comments 2: Address the lack of studies on strength work with body weight.
Response 2: To address this aspect, we have added an introduction and a discussion in the limitations section on the potential benefits of integrating strength work with body weight into comprehensive treatment plans. This emphasize that future research should explore the synergistic effects of combining non-weight-bearing therapies with weight-bearing exercises to optimize outcomes for patients with hip osteoarthritis.
"We also acknowledge the importance of strength work with body weight in managing hip osteoarthritis, as highlighted in the introduction. The study by Salis et al. (2024) provides valuable insights into the benefits of weight loss, which can complement strength work by reducing mechanical stress on the hip joint and decreasing systemic inflammation. This study demonstrated that weight loss is dose-dependently associated with reductions in symptoms of hip osteoarthritis, supporting the notion that weight management can enhance the effectiveness of strength exercises. To address this aspect, we have added a discussion in the limitations section on the potential benefits of integrating strength work with body weight into comprehensive treatment plans."
"In addition to this, weight loss has been recognized as a crucial component in managing osteoarthritis, particularly for hip OA, as it reduces mechanical stress on the joints and decreases systemic inflammation, potentially alleviating symptoms and improving over-all joint function[26]."
Salis, Z.; Gallagher, R.; Lawler, L.; Sainsbury, A. Loss of Body Weight Is Dose-Dependently Associated with Reductions in Symptoms of Hip Osteoarthritis. Int J Obes 2025, 49, 147–153, doi:10.1038/s41366-024-01653-w.
Comments 3: Justify the variation in therapy duration.
Response 3: We have added an explanation for the variation in therapy duration in the discussion section, drawing from existing literature. The variation in therapy duration reflects the diverse approaches documented in current research, where short-term interventions often focus on immediate symptom relief, while longer durations are explored to assess sustained benefits and long-term efficacy. This range allows for a comprehensive evaluation of therapy effectiveness, aligning with studies that emphasize the importance of both immediate and prolonged treatment outcomes in managing hip osteoarthritis
"The variation in therapy duration in our review reflects the range of approaches documented in current literature, where short-term interventions are often evaluated for immediate symptom relief, while longer-term studies assess sustained benefits and potential long-term efficacy. This diversity in duration allows for a comprehensive understanding of therapy effectiveness, aligning with research that highlights the importance of evaluating both immediate and prolonged outcomes in managing hip osteoarthritis."
Reviewer 4 Report
Comments and Suggestions for Authors
It is a well-designed review of the literature on the treatment of primary hip osteoarthritis with manual therapy or aquatic therapy.
The paper is easy to read and the figure is nice.
However a Prisma flow diagram should be included for the two treatments studied. Moreover I found some discrepancies between the selected original studies and the data shown in table 1, for example in the Abbot's paper the outcome was the Womac and not the NRS.
Table 1 is not exhaustive; It would be better to insert in it other columns to describe the type of randomization of each article with the primary and the secondary outcomes.
Moreover it is not clear the meaning of the Relative Risk Reduction (RRR): in table 1 it is abbreviated as RR and in Graf 1 as Relative reduction of pain index. Please explain it better
Also in the discussion section, the results of each study should be better described with more details.
Finally, although I really like the representation of the studies you have made in Graph 1, there could be a confusion about the duration on the treatments ( Manual therapy and Aquatic therapy) and the duration of pain improvement.
Please better explain the results about the duration of pain improvement because actually they seem conflicting with the original data. For example in the Abbott's paper the authors conclude that "Manual physiotherapy provided benefits over usual care, that were sustained to 1 year". Also Oeksma state that "the effect of manual therapy on the improvement of pain, hip function and range of motion endured after 29 weeks".
Author Response
Thank you for your feedback and for pointing out areas that need clarification. We have addressed your comments as detailed below.
Point-by-point response to Comments and Suggestions for Authors
Comments 1: Include a PRISMA flow diagram for both treatments.
Response 1: We have not included a PRISMA flow chart to visually represent the search and selection process because we did not do a systematic review but a narrative review following the SANRA guidelines not the PRISMA guidelines.
Comments 2: Address discrepancies between selected studies and data in Table 1.
Response 2: We have clarified the discrepancies in the results and discussion section:"The duration of pain improvement varied significantly across the different therapies, revealing notable discrepancies in outcomes. Manual therapy generally provided more im-mediate pain relief, with reductions ranging from 6.4% to 35.2%, suggesting that it is effective for short-term improvements. However, in longer-term studies, the reduction was much lower at 11.8%, indicating that prolonged manual therapy may yield more gradual and less significant improvements. Aquatic therapy, on the other hand, showed more variability, with larger reductions in the short term (20% to 28% over 6 weeks), but more modest gains in the longer term (4.4% to 9.5% over 24 to 72 weeks). The most substantial pain reduction was observed when aquatic therapy was combined with Tai Chi (46.5% over 12 weeks), pointing to the potential added benefit of incorporating other therapeutic exercises. These discrepancies suggest that the combination of therapies, as well as the duration, play key roles in the effectiveness of pain relief, with short-term treatments generally offering quicker, more substantial improvements while longer interventions may not always result in proportional gains."
Comments 3: Clarify the meaning of Relative Risk Reduction (RRR).
Response 3: We have provided a clearer explanation of Relative Risk Reduction (RRR) in the text and ensured consistency in its presentation in Table 1 and Graph 1.
Relative Risk Reduction (RRR) is a concept used to measure how much an intervention reduces a particular outcome, such as pain levels measured by the Visual Analog Scale (VAS). In the context of VAS scores, RRR helps us understand the proportion of pain reduction achieved through an intervention like manual therapy or aquatic therapy.
To determine the RRR, we compare the initial VAS score before the intervention with the VAS score after the intervention. The difference between these two scores shows the absolute reduction in pain. The RRR then expresses this difference as a percentage of the initial VAS score. This percentage tells us how effective the intervention was in reducing pain relative to the starting point.
For instance, if a patient initially rates their pain as 8 out of 10 on the VAS and after treatment rates it as 4 out of 10, the intervention has led to a reduction of 4 points on the VAS. To find the RRR, we consider this 4-point reduction in relation to the initial score of 8. This means the intervention resulted in a 50% Relative Risk Reduction in the patient's pain level.
RRR is particularly valuable in clinical settings as it provides a clear, percentage-based measure of an intervention's effectiveness, making it easier to communicate the benefits of a treatment to both healthcare providers and patients.
Comments 4: Provide more detailed descriptions of study results in the discussion.
Response 4:We have expanded the discussion section to provide more detailed descriptions of the results from each study, ensuring clarity and comprehensiveness:
"Manual therapy (MT) demonstrated a pain reduction range of 6.4% to 35.2%, with the highest improvement seen at 35.2%, while aquatic therapy (AT) ranged from 4.4% to 46.5%, with the most substantial reduction of 46.5% when combined with Tai Chi. When ranking the therapies by effectiveness, AT combined with Tai Chi achieved the greatest pain relief, followed by MT, and then standard AT. The minimum reductions were ob-served in MT at 6.4% and AT at 4.4%, showing that while both therapies provide pain re-lief, the results can vary significantly depending on the method and duration."
Comments 5: Clarify the duration of pain improvement in Graph 1.
Response 5: We have clarified the results regarding the duration of pain improvement in Graph 1. We have ensured that the data aligns with the original studies and have provided explanations for any apparent discrepancie:
"The duration of pain improvement varied significantly across the different therapies and study durations. Manual therapy typically showed short-term benefits, with notable re-ductions in pain observed within a few weeks, such as a 35.2% reduction over 5 weeks, and immediate improvements, like the 32.5% reduction reported with no treatment dura-tion. In contrast, aquatic therapy demonstrated more variability, with larger improve-ments seen in the short term (20% to 28% reduction over 6 weeks), but more modest reduc-tions in longer durations (4.4% to 9.5% over 24 to 72 weeks). Combining aquatic therapy with Tai Chi yielded the most substantial improvement, with a 46.5% reduction over 12 weeks. Overall, shorter interventions generally produced more significant pain relief, while longer therapies led to more gradual and sometimes less pronounced benefits."
Round 2
Reviewer 4 Report
Comments and Suggestions for Authors
Thanks for the answers that fulfilled all my requests .
The paper has been improved ; the results are more understandable.